# Meta-Learning Contextual Time Series Forecasting with Neural Processes

## Abstract

Neural Processes (NPs) are a powerful class of meta-learning models that can be applied to time series forecasting by formalizing it as a probabilistic regression problem. However, conventional NPs base their predictions only on observations from a single time series, which limits their ability to leverage varied contextual information. In this paper, we introduce a novel NP architecture that, in the spirit of meta-learning, is designed to incorporate context information from multiple related time series. To this end, our approach treats related time series as conditionally independent context examples of a shared underlying data-generating process corresponding to a specific meta-task. A sequence encoder aggregates a variable number of such context time series into a latent task description, which then conditions a sequence decoder, enabling accurate forecasting of unseen target time series. We evaluate our approach on challenging time series forecasting problems, demonstrating that our architecture performs favorably compared to a range of competitor approaches.

## 1 Introduction

Time series forecasting, i.e., the prediction of future values based on a sequence of past observations, is a fundamental task in many real-world applications such as predicting energy consumption, product demand, or controlling physical systems (Tzelepi et al., 2023; Boese et al., 2017; Lim & Zohren, 2020). In practice, time series data often exhibits complex, non-linear dynamics governed by an underlying stochastic process, whose properties may generally vary between different time series. Under such conditions, making accurate and uncertainty-aware predictions is challenging but essential for robust decision-making (Bishop & Bishop, 2024; Hüllermeier & Waegeman, 2021).

Naive deep learning approaches often demand substantial amounts of task-specific training data and exhibit poor generalization when applied to unseen tasks Goodfellow et al. (2016); Bishop & Bishop (2024). For many real-world applications, however, training data is scarce, rendering standard deep learning unfeasible. Meta-learning offers a powerful paradigm to overcome this challenge (Schmidhuber, 1987; Thrun & Pratt, 1998; Huisman et al., 2021; Hospedales et al., 2022). The core goal of meta-learning is to leverage data across multiple related tasks in order to extract inductive biases about the underlying task structure. This enables data efficient adaption to novel tasks during inference time. Bayesian variants are particularly well-suited for this setting, since they provide a principled approach for handling the uncertainty that naturally arises from limited data, thereby enhancing the robustness of meta-learning models Grant et al. (2018); Yoon et al. (2018); Garnelo et al. (2018b); Volpp et al. (2023).

The Neural Process Family (NPF, Garnelo et al. (2018b;a)) is a prominent class of Bayesian meta-learning models that has been extensively studied in the literature. Neural Processes (NPs) combine the flexibility of a neural network (NN) based architecture with the ability to model Bayesian uncertainty in order to learn stochastic processes, i.e., distributions over functions, in a data-efficient manner. NPs can be utilized for a variety of different applications and, in particular, have been applied in the time series domain, in order to make forecast predictions (Gordon et al., 2020) or to model the temporal dynamics of physical systems (Volpp et al., 2021).

A common approach for applying NPs in a time series domain is to formalize a time series as a function that maps a time index to an element of the series and to treat forecasting as a probabilistic regression problem Gordon et al. (2020); Bruinsma et al. (2023). In this formulation, each meta-task

is defined by a single time series, compelling the model to learn to make forecast predictions solely based on a context set of past observations from that same series. However, for many practical learning problems there is usually a broader context in the form of multiple related time series available. Incorporating context information from multiple time series is crucial to achieve good performance, especially if historical data for the target time series is limited and suffers from high noise Flunkert et al. (2017); Iwata & Kumagai (2020); Lim et al. (2019).

In this paper, we study meta-learning for the time series forecasting problem within the Neural Process framework, employing modern sequence models. In particular, our proposed method tackles the meta-learning problem in a conceptually correct way by treating an entire time series as a single context example. We introduce a structured meta-learning setting where each meta-task is defined by a specific data-generating-process with each task's dataset comprising multiple time series realizations of that process. We proposed a novel NP architecture for this setting, with the core innovation being its ability to treat each entire time series as a conditionally independent example of the underlying stochastic process. This sequence-level context aggregation empowers the model to better capture temporal dynamics within a single time series and, to effectively aggregate task information over multiple context time series. Through rigorous evaluation on various challenging time series forecasting problems, we demonstrate that our model achieves significant improvements over standard Conditional Neural Process variants, particularly in scenarios demanding robust generalization from limited or noisy data.

## 2 RELATED WORK

Traditional statistical time series forecasting methods such as ARIMA (Box & Jenkins, 1968) and Exponential Smoothing (Gardner, 2006), typically make strong assumption about the statistical structure of the data and often require manual selection of model parameters. More recent deep learning approaches, including recurrent neural network and transformer-based models, have shown remarkable performance on various sequence tasks, clearly outperforming statical methods (Flunkert et al., 2017; Lim et al., 2019; Huo et al., 2022). However, a significant limitation for both these traditional and deep learning methods is their reliance on substantial amounts of historical data from the specific time series that should be forecasted.

Early works in the meta-learning domain focused on learning feature extraction methods from past observations. The extracted features are then used to select a suitable forecasting model or hyperparameter configuration, often employing classical machine learning techniques (Lemke & Gabrys, 2010; Talagala et al., 2018).

More recently, optimization based meta-learning methods, such as Model-Agnostic Meta-Learning (MAML, Finn et al. (2017)), have been adapted for time series forecasting. These methods learn an optimized initial-ization for recurrent neural networks (RNNs) or attention-based models that can be rapidly fine-tuned to new time series using only a limited amount of historical data (Pineda-Arango et al., 2021; Narwariya et al., 2020).

A prominent and highly relevant line of work on meta-learning for time series data centers around Neural Process (NP) based models (Garnelo et al., 2018a;b). Neural Processes are designed to learn a mapping from a set of context observations to a predictive distribution over target points, inherently handling variable context sizes. In the context of forecasting, they are typically provided with a set of historical data points from a time series and learn to produce a probabilistic forecast for future values (Gordon et al., 2020; Bruinsma et al., 2023).

Several advancements have been proposed to enhance the capabilities of standard NPs by introducing specific inductive biases about the spatio-temporal structure of the data into the model architecture. Attentive Neural Processes (ANPs, Kim et al. (2019)) introduce attention mechanisms to mitigate the underfitting often observed in standard NP-based models, allowing them to better capture complex relationships between context data and predictions. Convolutional Conditional Neural Processes (ConvCNPs, Gordon et al. (2020)) incorporate translation equivariance of the modeled data as an inductive bias. This is particularly beneficial for time series data, as time series often exhibit some form of periodicity or recurring patterns over time. By leveraging convolutional neural networks, ConvCNPs can efficiently process sequential data and have demonstrated strong performance on time series data. Sequential Neural Processes (Singh et al., 2019) capture temporal

correlations between a sequence of context sets by modeling a recurrent latent space, enabling them to handle stochastic processes evolving over time. Recurrent Attentive Neural Process (Qin et al., 2019) extend ANPs for sequential data by integrating an RNN-based encoder to model temporally changing stochastic processes.

## 3 BACKGROUND

In this section, we establish concepts and notations used throughout this paper.

**Stochastic processes and time-series.** A stochastic process is defined as a collection of random variables $\{X_t\}_T$ (with index set $T$) that map from a probability space $(\Omega, \mathcal{F}, \mathcal{P})$ to a common measurable space $(\mathcal{Y}, \Sigma)$ Karlin & Taylor (1975). For a fixed event $\omega \in \Omega$, the mapping $T \ni t \mapsto X_t(\omega)$ defines a function $f : T \to \mathcal{Y}$ from the index set $T$ to the measurable space $(\mathcal{Y}, \Sigma)$. Therefore, a stochastic process can alternatively be viewed as a single random variable with values in the space of measurable functions $\mathcal{M}(T, \mathcal{Y})$, i.e., as a distribution over functions $p(f)$. The outcome of a stochastic process is called a realization or sample function. If $T = \mathbb{N}$, we can interpret the indices of the stochastic process as time and call it a (discrete) temporal stochastic process. The realization $(x_1, x_2, \dots)$ of a temporal stochastic process is called a time series. In practice, we also call a consecutive sample $x_{k:l} := (x_k, \dots, x_l)$ from any finite marginal of the underlying stochastic process a time series.

**Meta-learning.** The goal of meta-learning, also called learning-to-learn, is to build a machine learning model that can efficiently adapt to an unseen task given only a small set of context examples from that task (Schmidhuber, 1987; Thrun & Pratt, 1998; Huisman et al., 2021; Hospedales et al., 2022). Formally, we consider a distribution $p(\mathcal{T})$ of meta-tasks, where we assume that the tasks have common structure. A specific meta-task can be defined as learning a prediction model for a function $\mathcal{T} : \mathcal{X} \to \mathcal{Y}$ given a training data set $\mathcal{D}_\tau = \{(x_i, y_i = \mathcal{T}(x_i)) \in \mathcal{X} \times \mathcal{Y}\}$. Therefore, each task constitutes a standard supervised learning problem. A meta-learning model defines a procedure that maps a set of context examples $\mathcal{D}_\tau^C$ to a prediction model for the corresponding task. The meta-learning model achieves this by extracting inductive biases about the structure of the meta-task distribution. Training a meta-learning model requires a collection of task-specific data sets $\mathcal{D} \equiv \{\mathcal{D}_{\tau_i} = \mathcal{D}_{\tau_i}^C \cup \mathcal{D}_{\tau_i}^T \mid \mathcal{T}_i \sim p(\mathcal{T})\}$, which is called a meta-data set. Each task data set is usually divided into a context set $\mathcal{D}_\tau^C$ and a target set $\mathcal{D}_\tau^T$. During meta-training, the meta-learning model is optimized to produce the best predictive model for each task, leveraging examples from the context set. The quality of the prediction models is evaluated on the corresponding target sets.

**Probabilistic time series forecasting.** We consider the problem of probabilistic time series forecasting. The goal is to estimate the predictive distribution over future values of a temporal stochastic process given a history of observed time series elements (Rangapuram et al., 2018; Flunkert et al., 2017). Formally, let $\{X_t\}_{t \in \mathbb{N}}$ denote the latent stochastic process of interest, and let $\{Y_t\}_{t \in \mathbb{N}}$ denote the corresponding observed process defined by an observation model $Y_t \sim p(Y_t \mid X_1, \dots, X_t)$. Given a sequence of past observations $(y_1, \dots, y_l)$, the goal is to model the conditional distribution over future trajectories $p(Y_{l+1}, \dots, Y_T \mid (y_1, \dots, y_l))$.

**The Neural Processes Family.** The Neural Process Family (NPF, Garnelo et al. (2018a;b)) is a class of neural network-based models designed to meta-learn stochastic processes. The goal of Neural Processes (NPs) is to model the posterior predictive distribution $p(y \mid x; \mathcal{D}_\tau^C)$ of a stochastic process $p(f)$ conditioned on observed data, i.e., on a set of function evaluations $\mathcal{D}_\tau^C = \{(x_i, y_i = f_\tau(x_i) \mid i \in [N_\tau^C]\}$, where $N_\tau^C$ denotes the number of context points of task $\tau$ and $[n] := \{1, \dots, n\}$.

NP models can be divided into two sub-families. Latent Neural Processes (LNPs, Garnelo et al. (2018b)) utilize a stochastic latent variable to parameterize expressive non-Gaussian predictive distributions. However, this comes at the cost of requiring approximations during training. Conditional Neural Processes (CNPs, Garnelo et al. (2018a)) replace the latent variable with a deterministic context encoding and model the predictive distribution as $p_\theta(y \mid x, r_\tau^C)$, where $r_\tau^C = \text{Enc}_\theta(\mathcal{D}_\tau^C)$ is a fixed-sized deterministic deep set encoding of the context Zaheer et al. (2017). In contrast to LNPs, the model parameters $\theta$ can be optimized by standard gradient ascent on the log-likelihood function

evaluated on the meta-dataset, i.e.,

$$\mathcal{L}(\{\mathcal{D}_l\}_L) = \sum_{l=1}^{L} \sum_{m=1}^{M_l} \log p_\theta\big(y_{m,l}^T \mid x_{m,l}^T, \mathrm{Enc}_\theta(\mathcal{D}_l^C)\big).$$

A significant limitation of standard CNPs arises from the form of the learned predictive distribution, which is factorized conditioned on the context set. This restriction becomes especially severe when the likelihood parameterization $p_\theta(y \mid x, r^C)$ (often an isotropic Gaussian distribution) is too simplistic. While the architecture proposed in this paper is compatible with both conditional and latent Neural Process formulations, we nevertheless adopt a Conditional Neural Process (CNP) approach in this work. This choice is motivated by the practical advantages of CNPs as they prioritize simpler, deterministic optimization over the ability to generate diverse, correlated function samples. Crucially, this often leads to superior empirical performance, particularly when employing complex likelihood models, as is the case in our probabilistic time series forecasting setup (Gordon et al., 2020; Foong et al., 2020; Bruinsma et al., 2021; Markou et al., 2021; 2022; Bruinsma et al., 2023).

**Bayesian Context Aggregation.** A key property of NPs is the ability to make predictions given a variable number of context points. This requires some form of a permutation invariant aggregation mechanism that maps a variable-size set of elements $\mathcal{D}^C = \{(x_i, y_i)\}$ to a fixed-size, learnable representation $\bar{r} = \mathrm{Enc}_\theta(\mathcal{D}^C)$, called a deep set encoding (Zaheer et al., 2017; Wagstaff et al., 2022). Volpp et al. (2021) propose Bayesian Aggregation (BA) as a more principled approach that treats the context aggregation as a Bayesian inference problem. To this end, BA defines a latent observation model with Gaussian likelihood $p(r_i \mid z) = \mathcal{N}(r_i \mid z, \sigma_{r_i}^2)$ and Gaussian prior $p(z) = \mathcal{N}(z \mid \mu_z, \sigma_z)$. The posterior $p(z \mid \{r_i\}) \propto p(z) \prod_i p(r_i \mid z, \sigma_{r_i}^2)$ is also Gaussian and, thus, can be computed in closed form. A neural network-based encoder learns to map context points to corresponding latent observation together with their variances $(r_i, \sigma_{r_i}^2) = \mathrm{enc}_\phi(x_i, y_i)$. This simplifies the model architecture by unifying context aggregation and latent parameter inference into a single step. Moreover, unlike traditional mean aggregation, BA can easily assign varying levels of importance to individual context points, resulting in better inference. Although Bayesian Aggregation maps the context set to a distribution in the latent space, it can still be applied within the CNP framework. To this end, the likelihood model is conditioned on the aggregated mean and variance parameters, rather than sampling from the corresponding distribution. This effectively optimizes the predictive distribution by matching the first two moments of the likelihood to the corresponding moments of the latent posterior. Volpp et al. (2021) refer to this approach as parameter-based likelihood optimization and demonstrate empirically that it is a powerful aggregation scheme for CNP-based architectures.

## 4 BAYESIAN META LEARNING IN THE TIME SERIES DOMAIN

**Motivation.** A common approach for applying NP-based models in a time series domain is to treat forecasting as a multi-task probabilistic regression problem (Gordon et al., 2020; Bruinsma et al., 2023). In this setting, a time series is formalized as a function $f : \mathbb{N} \to \mathbb{R}^d;\ t \mapsto x_t$ that maps a time index $t$ to a time-series element $x_t$. Consequently, a meta-task corresponds to making predictions for a specific time series and the data set of a task is comprised of individual time series elements (i.e., $\mathcal{D}_\tau = \{(t_i, x_{t_i}) \mid i \in [M_\tau]\}$, where $M_\tau$ denotes the number of data points of task $\tau$). A collection of such time series can be used as a meta-dataset to train an NP-based forecast model. This approach has two major issues in practice:

1. **Conditional independence of data points.** In the standard CNP formulation (Garnelo et al., 2018a), the context points as well as the target predictions are modeled as being conditionally independent. However, this is not consistent with the explicit temporal structure of time series. Without including temporal dependencies as inductive bias, causality has to be learned implicitly from the data at the cost of predictive performance.

2. **Only context information from previous time steps.** By formalizing time-series forecasting as a regression problem, the model can merely utilize context elements from the target time series itself (i.e., time series elements at previous time steps). An implicit but fundamental assumption is that meta-learning extracts sufficient inductive bias regarding

the structure of the underlying stochastic process, thereby enabling the extrapolation of a time series' future solely from past observations. However, it can be necessary to provide additional contextual information beyond historical observations to achieve accurate forecast predictions. This is especially the case if the history is short or observations are rather noisy. One particular option is to incorporate information from additional related time series.

The first issue has been addressed by specialized CNP variants such as Convolutional Conditional Neural Processes (Gordon et al., 2020), Gaussian Neural Processes (Bruinsma et al., 2021) or Autoregressive Conditional Neural Processes (Bruinsma et al., 2023) which incorporate inductive bias about the spatial or temporal structure of the data into the model architecture and model predictive correlations. However, none of these approaches can be readily applied to utilize context information from multiple time series. In this work, we address both issues by proposing an alternative Conditional Neural Process architecture specifically designed to operate on time series data.

**Problem Statement.** In order to derive our approach, we consider a more structured meta-learning definition for probabilistic time series forecasting. To this end, we assume that the time series are realizations of a hierarchical stochastic process and make a clear distinction between the low-level data-generating process of a specific time series and the high-level meta-task distribution:

1. The low-level **data-generating-process** is a (usually parameterized) stochastic process $\{X_t\}_{t \in \mathbb{N}}^\tau$ that produces concrete time-series realizations.

2. The **meta-task distribution** is a high-level distribution over data-generating-processes, e.g., a distribution $p(\tau)$ over the parameters of the low-level parameterized process.

A specific data-generating-process corresponds to a single meta-task, and the data set of that task comprises multiple realizations of the underlying process (i.e., $\mathcal{D}_\tau = \{(x_{1:T_m})_m^\tau \sim \{X_t\}_{t \in \mathbb{N}}^\tau \mid m \in [M_\tau]\}$). In this setting, the meta-learning objective is to acquire the ability to make accurate forecast predictions for any meta-task, conditioned on a context set of multiple realizations from the corresponding process.

**Model.** Akin to standard CNPs, we propose an MLP based encoder-decoder architecture to encode a variable sized set of context time series into a fixed sized latent representation, which conditions the decoder to make forecast predictions. In the following we describe specifics about both components.

**Context Aggregation (Encoder).** In our hierarchical setting, each context time series corresponds to an individual, conditionally independent context example of the meta-task. In addition to that, we may have access to a sequence of past observations from the target time series. For the purpose of latent parameter inference, this history can be treated as an additional context sequence. Therefore, analogously to standard NP formulations, the model has to be able to aggregate a variable-sized set of context time series in a permutation invariant way. However, each context time series can, in turn, have an arbitrary length. We employ a neural network-based causal sequence model (e.g., a recurrent neural network (Elman, 1990; Hochreiter & Schmidhuber, 1997) or a causal transformer based architecture (Vaswani et al., 2017)), which enables the architecture to encode time series of variable length into a fixed-sized context encoding. Moreover, using a causal encoder explicitly accounts for the temporal structure of time series. The individual time series encodings can be mapped to the latent context set representation using any deep set encoding mechanism Zaheer et al. (2017). However, we follow Volpp et al. (2021) and propose to apply Bayesian Aggregation.

$$\mathcal{N}(\mu_{z_\tau}, \sigma_{z_\tau}) = p(z_\tau \mid \{r_i\}_\tau) \ , \ \ (r_i, \sigma_{r_i}^2)_\tau = \text{seqenc}_\phi\big((x_{1:T_i})_i^\tau\big)$$

As described in Section 3, BA incorporates weighting of context examples based on learned confidence into the context aggregation. This can be particularly important in settings where the context time series exhibit heterogeneous lengths, corresponding to greatly varying amount of information provided about the underlying stochastic process.

**Likelihood Parameterization (Decoder).** The conditional likelihood of future time series elements depends on both the latent context encoding $r_\tau = (\mu_{z_\tau}, \sigma^2_{z_\tau})$ and on the sequence of past observations $x_{1:l} = (x_1, \ldots, x_l)$. We model this as a factorized Gaussian distribution, with mean and variance parameterized by a causal sequence model

$$p_\theta(y_{l:T} \mid x_{1:l}, \mu_{z_\tau}, \sigma^2_{z_\tau}) = \mathcal{N}(y_{l:T} \mid \mu_{l:T}, \mathrm{diag}(\sigma^2_{l:T})) \, , \quad (\mu_{l:T}, \sigma^2_{l:T}) = \mathrm{seqdec}_\theta(x_{1:l}, \mu_{z_\tau}, \sigma^2_{z_\tau}) \, .$$

Employing a sequence model enables forecasting from a variable-length history and to incorporate time dependent covariates or exogenous inputs. Furthermore, the Gaussian parameters $\mu_{l:T}$ and $\sigma^2_{l:T}$ are likewise produced as a sequence over a variable-length prediction horizon (i.e., $\dim(\mu_t) = \dim(\sigma_t) = d$ and $\dim(\mu_{l:T}) = \dim(\sigma_{l:T}) = d(T - l)$).

**Meta-Training.** The meta-training procedure for our proposed model is similar to that of standard NP variants Garnelo et al. (2018a;b). During a training step, the dataset for each task in a mini-batch is partitioned into a context and a target set. The size of the context set is sampled randomly and, in addition to that, we also sample a random length for each context time series individually. In this way, the model can learn to properly handle task uncertainty arising from a variable amount of context data. For the target time series, we may sample a variable-length subsequence as a history. The model is trained to make forecast predictions for a fixed-length time horizon.

**Sampling Correlated Sequence Predictions.** A key limitation of standard CNPs is that they can not be used to acquire correlated function samples due to the factorized predictive distribution (Garnelo et al., 2018a). To overcome this issue, Bruinsma et al. (2023) propose an autoregressive sampling scheme for CNPs to define interdependent, non-Gaussian predictive distributions by iteratively conditioning subsequent predictions on previously generated outputs. This is implemented by adding these outputs to the context set. A strength of our method is that this autoregressive rollout strategy can be straightforwardly adapted to our model's sequence decoder. To this end, a prediction for the next time step can be sample from the forecast horizon and subsequently fed into the decoder as a new element in the observation histoy.

## 5 EXPERIMENTAL EVALUATION

In this section, we present a comprehensive experimental evaluation comparing the performance of our proposed approach against a range of competitive baselines. Our evaluation encompasses both synthetic data generated by simulating complex dynamical systems and real-world data, to rigorously assess model capabilities across different data characteristics.

**Dynamics Modeling (Synthetic Data).** A primary objective of our experimental evaluation is to assess the ability of different models to infer underlying meta-tasks from context data. For this purpose, we evaluate our approach on the challenging task of modeling the temporal dynamics of complex systems. Concretely, the objective is to forecast the sequence of future system states given an initial state, and conditioned on a sequence of control inputs (actions), i.e., $(s_0, a_{0:T}) \mapsto s_{1:T+1}$. This problem is formulated as a meta-learning problem by considering systems with varying physical parameters. Each unique configuration of these parameter values defines a distinct meta-task. A context time series is given by a state-action trajectory $(s_t, a_t)_{1:T}$. Beyond the context encoding, the decoder is conditioned on the first state of the target trajectory (i. e., corresponding to a history of length one) and additionally receives the sequence of actions as covariate input. In a fully observable setting, accurate forecast predictions are theoretically attainable with complete knowledge of the true task parameters. Thus, this scenario provides a crucial benchmark for evaluating a model's ability to infer the underlying meta-task from the given context trajectories. To generate diverse and challenging data, we utilize trajectory data from various reinforcement learning environments within the MuJoCo Playground framework Zakka et al. (2025) and modify the physical parameters of the simulation. For each meta-task, we generate multiple trajectories with varying initial configurations and goal conditions to ensure a diverse set of trajectories. We inject noise into the trajectories by adding Gaussian noise to the actions passed to the RL-environment. Our evaluation includes a diverse set of tasks including robotic manipulation and locomotion tasks. Comprehensive details on our data generation methodology are provided in Appendix A.1.2.

**Forecasting on Real-World Data.** To assess the ability of different models to effectively utilize context data alongside historical data from the target time series in a more ambiguous setting, we utilize real-world data from the UCR Time Series Archive Dau et al. (2019). This extensive dataset comprises time series classification data from various domains. It is important to note that real-world data often does not possess a clearly defined meta-task structure or explicit task parameters like our synthetic data. In our experiments with the UCR archive, we define context data as time series belonging to the same class as the target time series. This means that, in contrast to our synthetic data, there is no explicit or meaningful task parameter to infer. Instead, models must learn to leverage the inherent similarities within a class to improve forecasting. Model performance is evaluated on tasks (i.e., time series classes) that have not been used during training, simulating a few-shot generalization scenario. Further details on our data preparation methodology and selected datasets from the UCR archive are provided in Appendix A.1.2.

**Baselines and Competitors.** We compare our approach against the following competitor methods:

- **Conditional Neural Process (CNP):** As discussed in section 4, a generic approach for applying CNPs to time series forecasting uses individual time-series elements $(t, x_t)$ from the target time series as context points, thereby discarding the causal structure. To enable a fair evaluation that is directly comparable with our proposed architecture, we consider two variants. The first **Element-based CNP** variant uses the entire set of individual time-series elements from the provided context time series. For the second variant we follow Volpp et al. (2021; 2023) and instead employ a **Transiton-based CNP** where the context set is instead given by the set of all transitions (i.e., $\{(s_t, a_t, s_{t+1})\}$). These variants allow the CNP to access the same multi-time series context data as our model. Crucially, considering transitions is viable for our specific application of dynamics modelling because the underlying data generation process is Markovian, and transitions are independent of the absolute time index. The CNP utilizes a standard MLP-based encoder architecture with Bayesian Context Aggregation (Volpp et al., 2021).

- **Transformer-Concat:** This variant explores an alternative approach to context aggregation. All context sequences are concatenated into a single, long sequence. A transformer-based encoder then processes this concatenated sequence, implicitly learning a latent task representation. To distinguish individual trajectories within the concatenated input, a dedicated feature dimension is added to indicate the start of a new sequence. This setup effectively performs context aggregation through the self-attention mechanism of the transformer, without a separate, explicit aggregation mechanism present in NP-based models.

- **Oracle Model:** The Oracle model is explicitly conditioned on the true meta-task parameters (i.e., the varied physical parameters of the simulation used to generate the synthetic data). As such, it does not consider the context data to infer the meta-task. This baseline represents the upper bound of performance given full knowledge about the task and serves to quantify the irreducible error due to environmental stochasticity and data noise.

- **Uninformed Model:** In contrast to the Oracle, the uninformed model is context-oblivious. It makes predictions without access to any information about the meta-task, neither from context time series nor explicit task parameters. This baseline establishes the lower bound of performance, demonstrating the challenge of prediction without meta-task awareness.

For all evaluated models, including our proposed approach and all baselines, we employ the *same decoder architecture* based on a causal transformer. This ensures a fair comparison in which the performance differences can be primarily attributed to the context encoding and aggregation mechanisms rather than variations in the sequence generation capabilities of the decoder.

**Training and Evaluation Procedure.** The training procedure for our model is described in Section 4. We apply this identical procedure to all baselines, with specific adaptations for their respective architectures. For the CNP-based variants, sampled context time series are decomposed into individual elements or transitions, respectively. The Concatenated Context Transformer, conversely, processes these trajectories as a single, concatenated sequence. The Oracle and Uninformed models, by definition, operate without context encoding.

For each synthetic benchmark, the training dataset consists of 128 distinct meta-tasks, each comprising 32 trajectories, with each trajectory having a length of 64 timesteps. All models are trained to

forecast a prediction horizon of 32 timesteps. During both training and evaluation, the length of the context time series is randomly sampled between a minimum of 4 and a maximum of 32 timesteps. We evaluate performance on an unseen test set of identical size and structure to the training data. Crucially, this test set, including the sampled context and target sets, is fixed across all baselines and random seeds to ensure a fair comparison. For the real-world benchmarks, dataset sizes vary significantly across domains (details in Appendix A.1.2). Crucially, the number of tasks remains generally small, with a maximum of 32 training tasks observed across all domains.

Predictive performance is quantified using two primary metrics: the predictive log-likelihood and the mean-squared-error (MSE) of the mean predictions. Furthermore, we aim to evaluate a model's ability to infer the true meta-task parameters from the provided context. Note that this is only viable for the synthetic datasets. For this purpose, we separately train a simple MLP-based regression model to predict task parameters from the latent context encodings produced by a model. Note, that we do not propagate gradient from the regression model to the encoder. This ensures, that the model can not exploit usually unknown information during training. This regression model is trained on the MSE, but we show the $R^2$-coefficient for better comparability between different benchmarks. For all metrics, we report the mean and standard deviation across 7 independent experimental runs, each initialized with a different random seed.

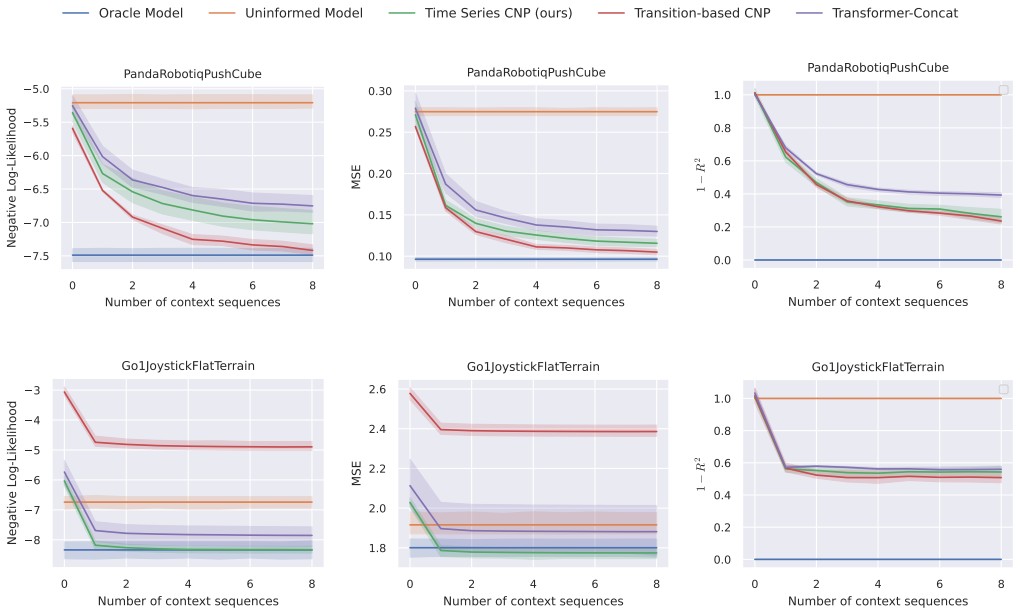

Figure 1: Evaluation of model performance on two dynamics modeling task with variable context set sizes. Our approach performs better than Transformer-Concat, both in terms of predictive performance (measured by neg. log-likelihood (left panel) and MSE (middle panel)) and in its ability to infer the task parameters (right panel). On the PandaRobotiqPushCube manipulation task traditional Transition-based CNP exhibits even stronger predictive performance while having a similar parameter inference accuracy. In contrast, on the Go1JoystickFlatTerrain locomotion task, the standard CNP struggles to produce meaningful predictions and is outperformed even by the uninformed baseline.

**Experimental Results for Dynamics Modeling (Synthetic Data).** On the box push manipulation task, all contextual model variants consistently exhibit performance metrics (both predictive log-likelihood and Mean Squared Error (MSE)) that fall between the uninformed and oracle baselines (see Figure 1 and Table 1). Without any contextual information (i.e., context set size 0), all variants show similar performance as the uninformed model. Crucially, performance consistently and significantly improves with an increased number of context sequences, demonstrating the value of contextual data from multiple independent context time series. Our proposed approach strictly outperforms the Transformer-Concat variant, across both predictive performance metrics. More importantly, our model demonstrates a superior ability to infer the underlying task parameters (i.e., box

| | Time Series CNP (ours) | | Transition-based CNP | | TransformerConcat | |
|---|---|---|---|---|---|---|
| | NLL | MSE | NLL | MSE | NLL | MSE |
| PandaRobotiqPushCube | $-6.62 \pm 0.55$ | $0.14 \pm 0.05$ | $\mathbf{-6.97 \pm 0.57}$ | $\mathbf{0.13 \pm 0.05}$ | $-6.39 \pm 0.49$ | $0.16 \pm 0.05$ |
| Go1 | $\mathbf{-8.05 \pm 0.73}$ | $\mathbf{1.81 \pm 0.08}$ | $-4.66 \pm 0.60$ | $2.41 \pm 0.07$ | $-7.58 \pm 0.74$ | $1.91 \pm 0.14$ |
| BerkeleyHumanoid | $\mathbf{-7.35 \pm 0.81}$ | $\mathbf{2.95 \pm 0.13}$ | $-4.65 \pm 0.96$ | $3.53 \pm 0.16$ | $-7.08 \pm 0.87$ | $3.00 \pm 0.16$ |
| T1 | $-23.54 \pm 1.61$ | $\mathbf{3.84 \pm 0.24}$ | $-12.27 \pm 1.68$ | $6.16 \pm 0.30$ | $\mathbf{-23.94 \pm 2.30}$ | $3.86 \pm 0.26$ |

Table 1: Negative log-likelihood (NLL) and mean squared error (MSE) for the contextual model variants averaged over all context set sizes. Our proposed model architecture performs best or similar to the competitors on all benchmarks. On the locomotion tasks, the Transition-based CNP exhibits significantly worse predictive performance compared to the other approaches.

mass and friction coefficients). This enhanced parameter inference, attributed to a more principled context aggregation, is a key factor contributing to the overall superior predictive performance of CNP-based methods compared to the purely transformer-based variant. Interestingly, the standard Transition-based CNP demonstrates even stronger predictive performance than our approach on this task, despite exhibiting similar parameter inference accuracy to our method. One explanation is that the model is not required to consider complex temporal dynamics since the box does not possess a complex state. Consequently, predicting its trajectory becomes a rather stationary task.

Results for one of the locomotion tasks are depicted in Figure 1 and Table 1. Further results are provided in the Appendix (Figures 3 and 4). On the locomotion tasks, our approach again consistently outperforms Transformer-Concat, albeit by a smaller margin. A significant advantage, however, is the notably reduced variance in performance across different random seeds, indicating a more stable model performance. Intriguingly, our model achieves performance close to the oracle with just a single context sequence, resulting in only marginal improvements when conditioning on additional sequences. We hypothesize this originates from the nature of the locomotion task's parameters. Here, the task parameters (i.e., the mass and static friction of the robot itself) directly influence the robot's trajectory at each time step. Thus, a single observed trajectory already provides rich information about the robot properties.

In contrast, for the box push manipulation task, the task parameters affect external object properties (i.e., the mass and friction of the pushed box) while the robot's own dynamics remain constant. Inferring these external properties requires observing multiple interactions between the robot and the box to build a comprehensive understanding of their behavior. In contrast to the box-push manipulation task, the standard CNP performs significantly worse compared to all other baselines, including the uninformed model. This poor predictive quality is evident in both predictive likelihood and mean predictions (MSE). Despite this, the standard CNP still demonstrates some utilization of the provided context, as its predictive performance significantly improves given at least one context sequence compared to no context at all. Furthermore, its parameter inference capabilities are comparable to our method and Transformer-Concat, yielding a similar $R^2$-coefficient for the inferred task parameters. It is noteworthy that parameter inference quality does not substantially improve beyond a single context sequence and a similar level of inference error consistently remained across all methods. Nevertheless, the drastic variation in predictive performance across models, despite comparable underlying parameter inference, is a critical observation. This strongly suggests that by inherently disregarding the explicit sequential structure within the context trajectories, the standard CNP struggles to robustly model temporal dependencies required to make accurate predictions. This architectural limitation likely leads to overconfident yet less accurate predictions, culminating in its significantly degraded predictive performance in locomotion tasks, even when its latent context embedding effectively captures the underlying task parameters.

**Experimental Results for Forecasting on Real-World Data.** On the Crop dataset, all contextual model variants significantly outperform the uninformed baseline, which relies solely on the target time series' historical data (see Figure 2). This stark performance difference underscores the value of leveraging contextual information for enhanced predictive accuracy. As anticipated, increasing the length of the provided history generally improves the predictive quality for both the uninformed baseline and contextual models when no additional context is explicitly used. However, the performance of models utilizing provided context largely stabilizes, indicating that context time series effectively capture critical information that might otherwise require longer historical windows. This suggests that contextual data can indeed be exploited to achieve more precise forecasting, even with

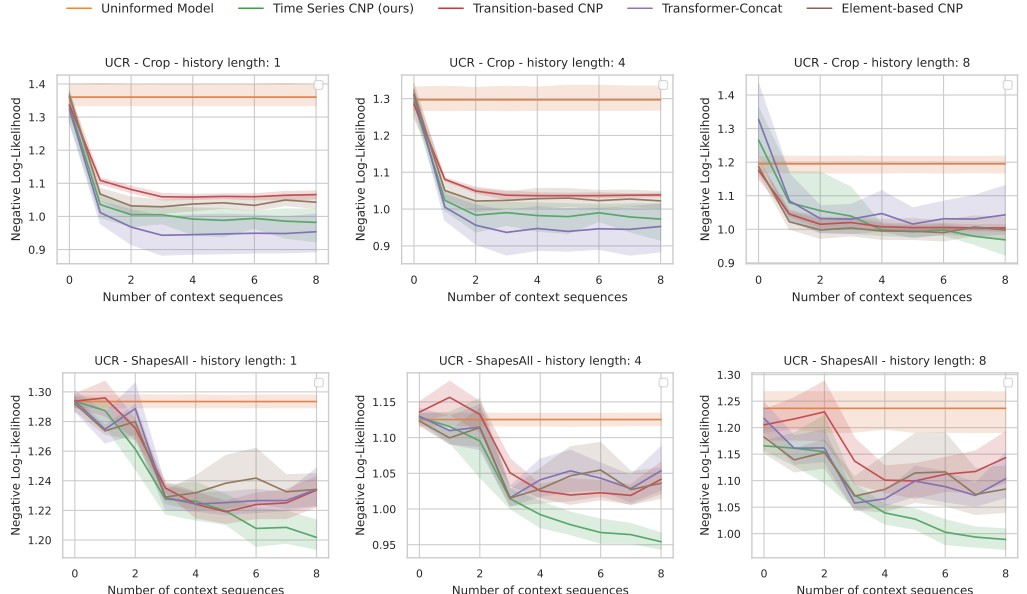

Figure 2: Evaluation of model performance measured by neg. log-likelihood on the UCR benchmark tasks with variable context set sizes and history length.

shorter target histories. Furthermore, our proposed method and the Transformer-Concat baseline consistently and significantly outperform both CNP baselines, highlighting their superior ability to model intricate temporal dependencies within contextual data.

For the NonInvasiveFetalECGThorax dataset, contextual models consistently achieve superior performance compared to the uninformed baseline, albeit the margin of improvement is comparatively smaller. Notably, predictive performance across all models does not show significant improvement with an increased length of the provided history. This observation suggests that for this specific domain, that the inherent data variance limits the utility of histories and context. Detailed results, are provided in the Appendix (Figure 5).

On the ShapesAll dataset, we once again observe substantial performance gains when incorporating additional context time series (see Figure 2). A particularly noteworthy finding is that our proposed method is uniquely capable of effectively utilizing more than three context time series. In contrast, all other evaluated contextual methods exhibit no further improvement in predictions when provided with more than three context examples. This superior capacity to leverage larger sets of contextual data points positions our method as particularly advantageous for scenarios where rich contextual information is available.

Overall, our proposed Time Series CNP architecture provides accurate predictions over the whole range of evaluated meta-modeling tasks, performing best or comparable to the competitor methods.

## 6 CONCLUSION AND OUTLOOK

We introduced a novel meta-learning approach for robust time series forecasting, leveraging a Conditional Neural Process (CNP) architecture designed to integrate context from multiple related time series. Our experiments demonstrated that while standard CNP architectures can achieve excellent predictions in scenarios with relatively static temporal dynamics, their limitations become apparent when accurate forecasting demands a deeper understanding of sequential dependencies. In such cases, achieving precise future predictions necessitates architectural inductive biases that explicitly model temporal structure. Furthermore, our findings underscore the importance of principled context aggregation within a meta-learning setting, resulting in superior and more robust performance.

REPRODUCIBILITY STATEMENT

To ensure a fair and rigorous comparison, we employed a consistent and robust training and evaluation procedure across all proposed and baseline models. Performance was assessed on a fixed test set, as detailed in Section 5, yielding statistically reliable and reproducible results. Hyperparameter settings and architectural specifics for all experiments are thoroughly documented in Appendix A.1.1. Furthermore, a comprehensive description of our data generation methodology for both training and testing sets is provided in Appendix A.1.2. We commit to releasing the full source code for this work with the camera-ready version of this paper to facilitate complete reproducibility.

ETHICS STATEMENT

This work focuses on fundamental research in machine learning methodology and does not involve human subjects, sensitive data, or applications with immediate societal impact. We do not expect any direct ethical concerns arising from this research.

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

# A APPENDIX

## A.1 EXPERIMENTAL DETAILS

### A.1.1 MODEL ARCHITECTURE AND HYPERPARAMETER

**Decoder Architecture.** To ensure a fair comparison, all baselines and contextual models tested in our experiments utilize an identical decoder architecture and architectural differences are confined solely to the encoder. The decoder architecture is concretely based on a causal transformer. The decoder's input is first projected to the transformer's embedding dimension via an MLP-based token embedding layer. We employ distinct embedding networks for the latent context encoding and for inputs related to the target time-series, specifically the start-state and the action sequence of the target trajectory. The transformer's output token is then mapped to the parameters (mean and variance) of a Gaussian likelihood function using two separate MLPs, one dedicated to predicting the mean and the other for the variance. Table 2 contains the decoder hyperparameters used in our experiments.

Table 2: Decoder Hyperparameters

| Parameter | Value |
|---|---|
| **General:** | |
| Activation Function | ReLU |
| Optimizer | AdamW (Kingma & Ba, 2015) |
| Learning Rate $\eta$ | $1 \times 10^{-4}$ |
| Batch Size | 16 meta-task (note, that the context set size is variable) |
| **Token embedding net:** | |
| Number of hidden layers | 2 |
| Hidden dimension | 256 |
| **Output nets for mean and variance:** | |
| Number of hidden layers | 2 |
| Hidden dimension | 256 |
| **Causal Transformer:** | |
| Number of layers | 2 |
| Embedding dimension | 256 |
| Number of attention heads | 8 |
| Feedforward dimension | 2048 |
| Dropout rate | 0.1 |
| Positional Encoding | sinusoidal |

**Encoder Architecture.** The specific encoder architectures for the contextual models evaluated by us are designed based on their respective approaches to handling the context sequences.

- **Transition-based CNP:** This model employs a simple MLP-based encoder to map individual transitions to latent observations.

- **Our Approach:** Our model utilizes a causal transformer-based sequence model to encode entire context time-series.

- **Transformer-concat:** This encoder directly aggregates concatenated context sequences into a latent representation by employing a transformer without causal masking, thereby allowing elements within different context sequences to attend to each other.

For both the Transition-based CNP and our proposed model, the encoders are designed to output the mean and variance of latent observations. These are subsequently aggregated using Bayesian aggregation (Volpp et al., 2021). Similar to the output nets of the decoder, we use separate MLPs to learn mean and variance of the latent observations. The transformer architectures for the encoders of our model and Transformer-concat are symmetric to the decoder architecture. The specific hyperparameters for the encoder architecture of the Transition-based CNP are given in Table. 3.

Table 3: Encoder Hyperparameters

| Parameter | Value |
|---|---|
| **General:** | |
| Activation Function | ReLU |
| Optimizer | AdamW (Kingma & Ba, 2015) |
| Learning Rate $\eta$ | $1 \times 10^{-4}$ |
| Batch Size | 16 meta-task (the context set size is variable) |
| Latent context embedding dimension | 128 |
| **Transition-based CNP:** | |
| Number of hidden layers | 2 |
| Hidden dimension | 256 |

### A.1.2 DATASET GENERATION

**Dynamics Modeling (Synthetic Data).** To generate trajectory data for our experiments, we utilize MuJoCo Playground (Zakka et al., 2025) environments. We create a set of meta-tasks by systematically varying the physical property of the underlying simulations. Details on the meta-task parameters are provided in Table 4. All parameter values are scaling factors multiplied with the default value of the respective property.

Our data generation process begins by training a reinforcement learning policy on the default environment (i.e., without parameter modifications). For all environments, we employ a PPO-based policy (Schulman et al., 2017) configured with the standard settings provided by the MuJoCo Playground framework. Once trained, this policy is repeatedly rolled out in the default environment to produce a large collection of reference trajectories. Subsequently, we sample task parameters to define our meta-tasks. For each meta-task, we generate a fixed number of trajectories using a two-step process: First, we randomly select an equal number of previously generated reference trajectories. Second, we execute the action sequences from these sampled reference trajectories in the corresponding meta-task environment (with varied parameters), always starting from the same initial state as the original reference trajectory.

We deliberately chose this open-loop trajectory generation method over directly rolling out the policy in a closed-loop fashion within the meta-environments. Closed-loop policy rollouts tend to steer trajectories towards a mean, simplifying the forecasting problem. Furthermore, conditioning a sequence decoder on task-specific actions in a closed-loop setting can lead to meta-overfitting (Rajendran et al., 2020; Yin et al., 2020). During trajectory generation, we disable the standard observation noise present in MuJoCo Playground. Instead, we introduce noise by executing noisy actions within the meta-environments, resulting in a challenging scenario where the effective noise increases with trajectory length.

| Environment | Varied environment parameters | Rarameter distribution | std. of injected action noise |
|---|---|---|---|
| PandaRobotiqPushCube | box mass
box friction coeff. | $\mathcal{U}(0.5, 2.0)$
$\mathcal{U}(0.5, 2.0)$ | 0.025 |
| Go1JoystickFlatTerrain | mass of robot links
static joint friction | $\mathcal{U}(0.7, 1.3)$
$\mathcal{U}(0.7, 1.3)$ | 0.025 |
| BerkeleyHumanoidJoystickFlatTerrain | mass of robot links
static joint friction | $\mathcal{U}(0.7, 1.3)$
$\mathcal{U}(0.7, 1.3)$ | 0.025 |
| T1JoystickFlatTerrain | mass of robot links
static joint friction | $\mathcal{U}(0.8, 1.2)$
$\mathcal{U}(0.8, 1.2)$ | 0.01 |

Table 4: Details on used meta-task parameters per environment.

**Real-World Forecasting Data.** The UCR Timeseries Archive (Dau et al., 2019) is a comprehensive collection of time series classification datasets sourced from diverse real-world domains. Each dataset features a varying number of classes, with multiple time series instances per class. Datasets are pre-split into distinct training and testing sets, each containing representatives from all classes.

For our experiments, we first merge these pre-defined training and testing sets. We then filter this merged collection, retaining only datasets that comprise at least 16 classes and 16 time series per class. To ensure consistency between different datasets, all time series in each dataset are uniformly subsampled (if needed) such that they do not exceed a maximum length of 128 steps.

Crucially, our methodology treats each class as an independent meta-task. Consequently, we re-partition each dataset into new training and testing sets based on classes: the training set contains a subset of all classes, while the test set comprises the remaining, unseen classes. The precise sizes of these class-based splits are adequately chosen and detailed in Table 5.

| Dataset | nr. of classes/tasks (train + test) | time series per task | time series length (subsampled) |
|---|---|---|---|
| Crop | 24 (16 + 8) | 32 | 46 |
| NonInvasiveFetalECGThorax | 42 (32 + 10) | 32 | 93 |
| ShapesAll | 60 (48 + 12) | 20 | 128 |

Table 5: Details on UCR real-world forecasting dataset.

### A.1.3 EVALUATION PROCEDURE

We evaluate the model performance on an unseen test set that is fixed across all tested models and random seeds. Crucially, this test set, also fixes the sampled context and target sets per meta-task. Given a specific context set, we evaluate the model on multiple target time series.

**Evaluation Metrics.** We use three different metrics to assess the predictive performance and the ability to infer the latent task parameters of a model.

- **Predictive Log-Likelihood**
- **Mean Squared Error**
- $R^2$**-coefficient:** In order to assess a model's ability to infer the task parameters $z_\tau$ from the provided context, we train a separate MLP-based regression model to maps the latent context encoding $r_\tau = \text{Enc}_\phi(\mathcal{D}_\tau^C)$ to the true task parameters $\hat{z}_\tau = \text{paramdec}(r_\tau)$. This regression model is trained with a standard MSE loss. However, since a MSE in a parameter space is hard to interpret and to compare between different benchmarks, we instead report the $R^2$-coefficient of the regression model, which is defined as

$$R^2 = 1 - \frac{\sum_\tau (z_\tau - \hat{z}_\tau)^2}{\sum_\tau (z_\tau - \overline{z})^2} \ .$$

  Here, $\overline{z}$ denotes the mean task parameters over all meta-tasks.

Log-Likelihood and mean squared error are computed for each target time series, averaging over all elements in the series. Finally, the metrics are averaged over all tasks and all target time series per task.

## A.2 ADDITIONAL EXPERIMENTAL RESULTS

This Section of the Appendix contains plots with experimental results for further benchmarks.

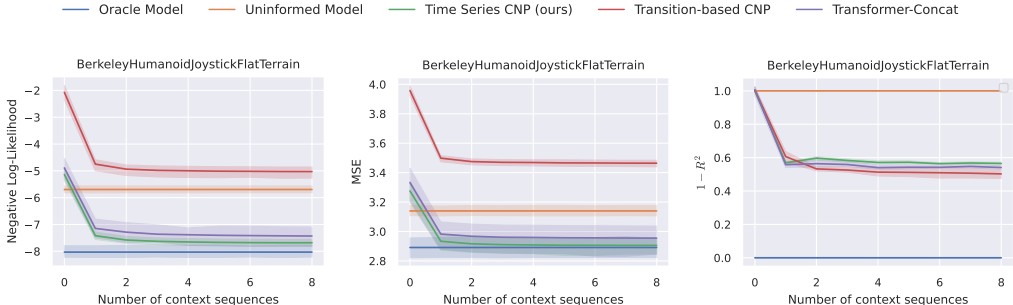

Figure 3: Evaluation of model performance on the BerkeleyHumanoidJoystickFlatTerrain locomotion task over variable context set size. Our approach performs better than Transformer-Concat, both in terms of predictive performance (measured by neg. log-likelihood and MSE) and in its ability to infer the task parameters. In contrast, the standard CNP's performance is inferior even to the uninformed baseline.

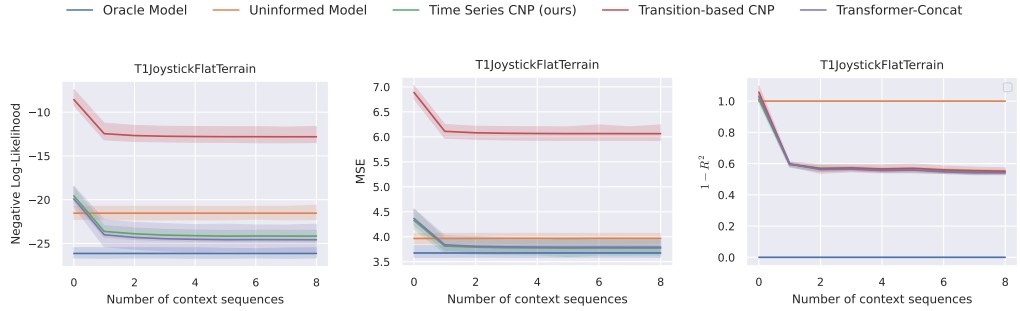

Figure 4: Evaluation of model performance on the T1JoystickFlatTerrain locomotion task over variable context set size. Our approach performs better than Transformer-Concat, both in terms of predictive performance (measured by neg. log-likelihood and MSE) and in its ability to infer the task parameters. In contrast, the standard CNP's performance is inferior even to the uninformed baseline.

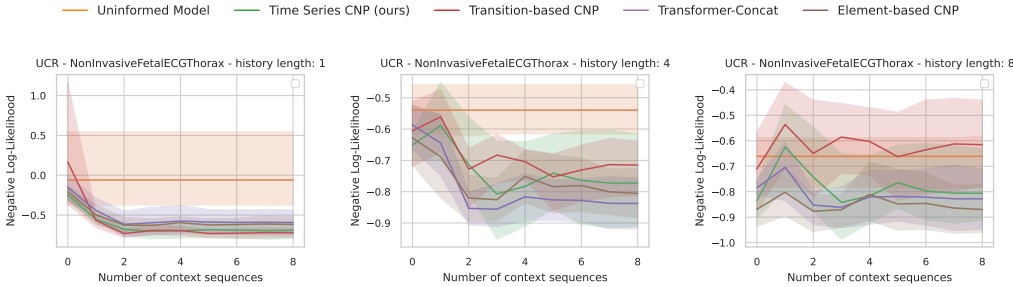

Figure 5: Evaluation of model performance measured by neg. log-likelihood on the UCR - NonInvasiveFetalECGThorax benchmark tasks for variable context set sizes and history length.

| | Time Series CNP (ours) $1 - R^2$ | Transition-based CNP $1 - R^2$ | TransformerConcat $1 - R^2$ |
|---|---|---|---|
| PandaRobotiqPushCube | $0.44 \pm 0.23$ | $\mathbf{0.43 \pm 0.24}$ | $0.52 \pm 0.19$ |
| Go1JoystickFlatTerrain | $0.60 \pm 0.15$ | $\mathbf{0.57 \pm 0.16}$ | $0.62 \pm 0.15$ |
| BerkeleyHumanoidJoystickFlatTerrain | $0.62 \pm 0.14$ | $\mathbf{0.58 \pm 0.16}$ | $0.60 \pm 0.15$ |
| T1JoystickFlatTerrain | $\mathbf{0.61 \pm 0.14}$ | $0.62 \pm 0.16$ | $0.61 \pm 0.15$ |

Table 6: Experimental results on the model's ability to infer the true meta-task parameters from the provided context averaged over all context set sizes.

