# OpenReview forum: "Meta-Learning Contextual Time Series Forecasting with Neural Processes"
_ICLR.cc/2026/Conference — Submitted to ICLR 2026_

### Official Review · Reviewer_oXKk · 2025-10-28

**Soundness:** 2
**Presentation:** 1
**Contribution:** 2
**Rating:** 2
**Confidence:** 2

**Summary:**

In this paper the authors propose a meta learning framework that utilizes Bayesian neural processes for time series forecasting with context examples. The key novelty is the use of Bayesian aggregation to merge the hidden representation of an indefinite amount of context examples into the formulation of conditional NPs. In the empirical study section the authors demonstrate the effectiveness of their proposal over a few ablation baselines on timestamped traces collected from RL environments.

**Strengths:**

The paper studies a practically meaningful question of time series forecasting with few-shot examples. The proposed method of bayesian context aggregation is a clean formulation in the bayesian NP framework.

**Weaknesses:**

While the overall theoretical idea is clean, the writing of the paper is rather hard to follow with many choices made in the paper not explained. The empirical study is also remotely convincing, that it lacks other baseline methods and the benchmark tasks are arguably not about time series. See questions.

**Questions:**

1. Regarding the theoretical perspective, is there equivalence between the Bayesian formulation and the method of MLP on top of the deep set aggregation of context example encodings concatenated with the representation of the task history?

2. Regarding writing:
(1) Some key details are missing in the main body: e.g., a clean formulation of the exact question and data generative distribution used in the study, the exact model architecture, a sketch of a forward pass, stepwise summary of the training algorithm, to name a few.
(2) The writing is not well organized in its current shape, which makes it quite hard to see the contributions of the paper. For example, section 1 through 4 are too dense touching topics not in order while have little information to distinguish the proposed method from existing ones.

3. The tasks involved in the empirical studies are new traces from dynamic systems generated by RL environment. In my humble opinion these are not the go to tasks when one thinks of time series forecasting which usually involves some degree of trend and seasonality on top of autocorrelation. Consider including other established time series benchmarks which would also help demonstrate the generalizability of the proposed method.

4. Similarly there are no other time series forecasting methods involved as baselines to justify the advantage of the proposed method. If the context examples are the limiting factor, at least consider testing in the uninformed manner, or apply multivariate models treating contexts as other variates after conditioning on the number of context examples.

---

> ### Author Response · Authors · 2025-11-28
> **Answer to reviewer oXKk**
>
> We thank the reviewer for their insightful comments and appreciate their recognition that our paper studies a
> "practically meaningful question of time series forecasting with few-shot examples" and presents a "clean theoretical idea"
> with Bayesian context aggregation.
>
> Below, we address the questions and weaknesses brought up by the reviewer:
>
> - Q1: Bayesian aggregation can be seen as a generalization of mean aggregation. It contains mean aggregation as a special
> case by assigning uniform observation variance to all context encodings (see [1], Section 4.1).
>
> - Q2: We are actively working on a revised version of our manuscript, to address the structural issues brought up
> by the reviewers. More concretely, we focus on 1.) shortening introduction and background, dedicating more space
> and clarity to our method and contributions upfront. 2.) Review and clarify all mathematical notations. 3.) Adding details
> related to the description of our method.
>
> - Q3: We deliberately considered dynamical system benchmarks for our evaluation, since these simulations offer precise
> control over data generation and a clear meta-task structure with known meta-parameters. This allowed us to precisely
> evaluate the model's capability to infer concrete meta-tasks, as demonstrated in the paper. Regarding noise, we did
> incorporate heteroscedastic noise to the time series by adding action noise during trajectory generation.
>
>   While using datasets with clearly defined meta-task structures is a common and accepted practice in the meta-learning/neural process literature [1,2,3], we agree with the reviewer that incorporating further datasets is a valuable contribution. We are addressing this by conducting experiments with time series data from the UCR time series archive [4]. This collection consists of real-world time series from various domains. We will incorporate these results and an in-depth discussion of the findings into the updated manuscript (the experiments are currently running).
>
>   We would also be very grateful if the reviewer could suggest any further real-world time series datasets or benchmarks  they believe would be particularly suitable for evaluating our method's performance and generalizability in practical settings.
>
> - Q4: Our initial approach focused on comparing with the classical CNP model because we introduce our method as an alternative
> CNP architecture, specifically designed to aggregate multiple full context time series in a principled way. We consider
> the standard CNP a relevant baseline for demonstrating the benefits of our aggregation mechanism. Our experiments show that
> incorporating information from multiple improves predictions compared to using an equivalent amount of data from a single series.
> Indeed, as the reviewer notes, a significant hurdle for applying many traditional time series forecasting models in our
> meta-learning setting is their inherent inability to directly incorporate multiple context time series as structured
> meta-information.
>
> [1] https://openreview.net/pdf?id=ufZN2-aehFa
>
> [2] https://arxiv.org/abs/1910.13556
>
> [3] https://arxiv.org/abs/2303.14468
>
> [4] https://arxiv.org/abs/1810.07758

---

### Official Review · Reviewer_Qe2b · 2025-10-29

**Soundness:** 3
**Presentation:** 2
**Contribution:** 3
**Rating:** 4
**Confidence:** 4

**Summary:**

This paper introduces a neural process-based model designed for meta-learning in probabilistic time series forecasting. Compared with previous works that make predictions only based on previous observations, this work incorporates context information from multiple related time series to form a meta task. A sequence encoder is used to encode context series into a latent representation. Multiple representations are aggregated via Bayesian context aggregation to condition a decoder to generate predictions. Experiments on simulated reinforcement trajectories validate the effectiveness of the proposed model.

**Strengths:**

1. The issue studied, leveraging multiple related time series for improved forecasting performance, is fascinating and highly relevant.
2. The proposed model is simple and easy to understand.

**Weaknesses:**

1. The presentation needs improvement:
  - Too much space is used to introduce the less critical background technical details, well, the core problem statement and methods only appear on page 5 and are very brief.
  - The notations used are too complicated, making readers confused.

2. Despite that, utilizing related series to improve forecasting is interesting; the result model that encodes each series using an encoder and aggregates representations with Bayesian context aggregation is a little bit trivial.

3. Experiments are only conducted on simulated MuJoCo state trajectories; broader experiments on real-world time series datasets are required.

4. Only some ablated versions of the proposed model are compared, lacking comparison with previous works.

5. An ablation study is required to validate the advantages of: 1)utilizing related series vs. predicting based only on past observation; 2) Bayesian context aggregation v.s. Simple aggregation like mean pooling.

**Questions:**

In real-world datasets other than controllable RL environments, how to construct a meta-task dataset with related series?

---

> ### Author Response · Authors · 2025-11-28
> **Answer to reviewer Qe2b**
>
> We thank the reviewer for their feedback and for acknowledging the "fascinating and highly relevant" nature of our
> problem setting. We also appreciate the positive comment that our "proposed model is simple and easy to understand."
>
> In the following we address the questions and weaknesses:
>
> - W1: We are actively working on a revised version of our manuscript, to address the structural issues brought up
> by the reviewers. More concretely, we focus on 1.) shortening introduction and background, dedicating more space
> and clarity to our method and contributions upfront. 2.) Review and clarify all mathematical notations. 3.) Adding details
> related to the description of our method.
>
> - W3/Q1: We deliberately considered dynamical system benchmarks for our evaluation, since these simulations offer precise
> control over data generation and a clear meta-task structure with known meta-parameters. This allowed us to precisely
> evaluate the model's capability to infer concrete meta-tasks, as demonstrated in the paper. Regarding noise, we did
> incorporate heteroscedastic noise to the time series by adding action noise during trajectory generation.
>
>    While using datasets with clearly defined meta-task structures is a common and accepted practice in the meta-learning/neural process literature [1,2,3], we agree with the reviewer that incorporating further datasets is a valuable contribution. We are addressing this by conducting experiments with time series data from the UCR time series archive [4]. This collection consists of real-world time series from various domains.  We will incorporate these results and an in-depth discussion of the findings into the updated manuscript (the experiments are currently running).
>
>    We would also be very grateful if the reviewer could suggest any further real-world time series datasets or benchmarks they believe would be particularly suitable for evaluating our method's performance and generalizability in practical settings.
>
> - W4: Our initial focus was on comparing with the standard CNP architecture because we position our work as a novel variant
> within this framework, specifically designed to aggregate multiple full context time series. We consider the standard
> CNP a relevant baseline for demonstrating the benefits of our aggregation mechanism, rather than an ablated version of our method.
>
>     To best address this weakness, we would be grateful if the reviewer could suggest specific previous works or types of models they would like to see included. We are open to considering other CNP-based baselines such as Attentive CNPs or Convolutional CNPs and other comparable time series forecasting methods. We acknowledge the challenge that many traditional forecasting models are not inherently designed to incorporate multiple context time series in a meta-learning fashion, as our method does.
>
> - W5.1: We have indeed conducted preliminary ablation studies demonstrating that aggregating data from multiple context
> time series improves predictions compared to using an equivalent amount of data from a single series
> (e.g., historical observations). We will incorporate these detailed ablation results and analysis into the revised manuscript.
>
> - W5.2: The original BA paper [1] did demonstrate its ability to effectively down-weight less relevant context data in contrast to
> more standard mean aggregation. However, we did not validate this specifically in our experiments. We consider incorporating
> specific ablation studies into the revised manuscript to empirically demonstrate how well Bayesian Aggregation handles
> irrelevant or out-of-distribution context sequences compared to mean aggregation.
>
> [1] https://openreview.net/pdf?id=ufZN2-aehFa
>
> [2] https://arxiv.org/abs/1910.13556
>
> [3] https://arxiv.org/abs/2303.14468
>
> [4] https://arxiv.org/abs/1810.07758

---

### Official Review · Reviewer_3dpa · 2025-10-31

**Soundness:** 2
**Presentation:** 1
**Contribution:** 2
**Rating:** 2
**Confidence:** 2

**Summary:**

This paper proposes a meta-learning approach for contextual time series forecasting using neural processes. The work introduces a neural processing framework for multivariate time series that employs in-context learning to address the problem of limited data and enable few-shot learning. The paper appears to present a Bayesian variant neural network that modifies standard neural processes to tackle meta-learning for multivariate time series. According to the abstract and introduction, a key motivation is addressing a limitation in existing methods that focus on individual time series forecasting rather than leveraging information from multivariate time series collectively. The authors propose alternative conditional neural process architectures specifically designed to operate on time series, potentially using recurrent neural networks or causal transformers.

**Strengths:**

## Strengths

- The third paragraph of the introduction provides a helpful overview of neural processes as neural networks designed to learn stochastic processes.
- The related work on neural processes suggests this is a promising research direction.
- The experimental results appear favorable based on the highlighted outcomes presented.

**Weaknesses:**

## Weaknesses

**Clarity and Presentation:**
- The core contribution and essential technique are unclear even after reading the introduction (lines 60-70). An enumerated list of contributions would significantly improve clarity.
- The presentation is inaccessible to readers outside this specific domain. The paper assumes too much specialized knowledge without providing sufficient intuitive explanation.
- The paper lacks concrete examples or running examples that would help readers understand the approach.
- Critically, there are no graphical illustrations of the proposed model, making it extremely difficult to understand the architecture and data flow.

**Technical Exposition:**
- By page 4, halfway through the paper, the proposed method remains unclear. The background material is overly introductory and could be shortened, yet it fails to build toward a clear understanding of the contribution.
- Section 4 (Method) is very difficult to follow. The equations are presented at a high level without clearly conveying the actual model architecture or implementation details.
- The relationship between mentioned components (e.g., recurrent neural networks or causal transformers in line 258) and the mathematical notation (e.g., equation in line 265) is unclear and too briefly explained.
- Where and how neural networks are incorporated into the neural process framework is never made explicit.

**Notation and Mathematical Rigor:**
- Notation inconsistencies cause confusion. For example, d_tc appears related to f_t (line 152), but f_t's definition is unclear, and the notation d_tc should possibly be d_ft given earlier definitions (line 134).
- Different symbols appear similar or are confusing (calligraphic T in line 154 vs. line 133, which turn out to be τ and calligraphic T respectively).
- The loss function is not clearly specified. It is unclear whether cross-entropy or log-likelihood is used.

**Logical Consistency:**
- Line 213 contains confusing or potentially contradictory statements about using univariate individual time series versus incorporating information from related time series at previous time steps.
- The Bayesian context aggregation section (around line 173) lacks clarity about what probabilistic quantities are being computed.

**Overall Impact:**
- After carefully reading through Section 4, I remained unable to understand the proposed method despite significant effort. This prevented meaningful evaluation of the experimental results.
- The paper does not meet the reasonable expectation that a reader with a PhD should be able to grasp at least the intuition behind the proposed approach.

**Recommendation:** The paper requires major revisions focusing on clarity of presentation, concrete examples, visual aids, explicit architectural details, and clearer mathematical exposition before it can be properly evaluated.

**Questions:**

I have listed structured questions (with help of LLM) in the above weakness part. I am going to say here my honest thoughts when reading the paper as it presents, and hopefully this can help you understand how a new reader perceives your paper. These raw feelings are genuine and I hope they provide a more human-to-human communication and contexts for the structured question above.

# Review: Meta Learning Contextual Time Series Forecasting with Neural Process (Paper 18560)

I am reviewing a paper on meta-learning contextual time series forecasting with neural processes. The paper appears to introduce a neural processing framework for multivariate time series. From the abstract, it seems to be using context in meta-learning, though the specific mechanism is unclear after this initial reading.

## Introduction

After reading the first two paragraphs of the introduction, I understand this is a meta-learning Bayesian variant neural network for few-shot learning. The third paragraph is helpful regarding neural processes, which are essentially neural networks designed to learn stochastic processes. Line 55 discusses a limitation in the formulation of existing methods that seem to focus on individual time series forecasting rather than leveraging multivariate time series collectively.

Lines 60 to 70 appear to describe the contribution, but the presentation is unclear. An enumerated list would help here. My vague intuition is that this involves meta-learning and some form of in-context learning. However, after reading this paragraph, I still do not understand the essential technique that achieves the proposed improvements.

## Related Work

Line 82 clearly implies that the method will employ in-context learning to solve the problem of limited data and enable few-shot learning. After reading the related work section, it is clear that the paper proposes some modification of neural processes to tackle meta-learning for multivariate time series and few-shot learning. The related follow-up work on neural processes makes this direction promising.

## Background (Section 3)

While I intend to read Section 3 in detail, glancing through the structure of the paper, I realize that up to page 3, I still do not know what the authors are actually proposing. The first two paragraphs on stochastic processes, time series, and meta-learning are very introductory and could be shortened. I still do not have a concrete understanding of what the authors are trying to accomplish.

After reading all of the background material, I am now up to page 4, halfway through the paper. The approach is still not crystal clear or intuitive to me. A running example would help significantly. After reading about the neural process family, I have a high-level understanding of the concept, but the roles of context and targets, and where the neural network comes into play, remain confusing. One reason for this confusion may be that the description has been very generic.

I find myself staring at certain parts for a long time. For instance, in line 152, there is a predictive distribution of d_τ or T (calligraphic T) of c. This d, denoted as d_tc, is related to f_t. However, where is that f_t defined? Earlier, there is a d_t notation in line 134, and I understand that t corresponds to the function mapping between y and x. Therefore, d_tc should really be d_ft. Additionally, I assume the calligraphic T in line 154 is different from the T in line 133 and has a very different meaning. Upon closer inspection, I see that in line 154, it is actually τ (tau), and in line 133, it is calligraphic T. (You are really pushing the limit of my eyes.)

After reading the neural process family section (lines 150-176), I still do not understand where the neural network comes into play. The section on Bayesian context aggregation leaves me very confused. My understanding is that it relates to a posterior distribution, but I am uncertain. As of line 173, having read the first three sections, I do not have a concrete example or concrete mathematics to understand what is exactly happening. The description has been high-level, but not in a way that builds intuition. I also lack intuition about the underlying mechanism.

## Method (Section 4)

In line 228, I encounter a sentence that concretely addresses the proposal: the authors are proposing alternative conditional neural process architectures specifically designed to operate on time series. However, I still do not know what this entails—the sentence feels hollow.

Line 213 is confusing. It seems to say that the method uses only a univariate individual time series from its own past, yet there might be other information from the past for other related time series. I would be very careful here. For related time series, the method should also use information from those related time series at previous time steps, which does not align well with the boldface sentence in line 213.

I find Section 4 very difficult to read. The equations are high-level and do not clearly convey the architecture or the actual model. All I can discern is a few-shot learning approach for a conditional distribution. Surprisingly, there is not a single graphical illustration of the model to aid understanding. Without an intuitive example or exact mathematical formulation, how am I supposed to understand what the authors are proposing?

For example, the authors mention using a recurrent neural network or a causal transformer in line 258. How does that relate to the notation in the equation at line 265? The explanation is too brief. Line 265 appears to already involve Bayesian aggregation, but I do not understand what is actually being done.

At line 270, I see that there is an output (y_hat) that is a Gaussian distribution, with the model outputting the mean and variance. This yields a normal distribution. What is the loss function? Cross-entropy? If the authors specify a log-likelihood parameterization decoder, are they using log-likelihood as the loss function?

Regarding meta-training in line 280, the authors state it is similar to that of standard neural process variants during the training step. At this point, I am simply lost. I have tried very hard to understand the method, but I cannot.

## Experiments and Results

I glanced through the experiments and results. Because I have so little understanding of the methods, I cannot properly evaluate the results. The outcomes appear favorable based on the boldface highlights, but there is not much I can meaningfully assess.

## Overall Assessment

Unfortunately, because I did not truly understand the method, I would not be able to rate this paper highly. I believe it is reasonable to expect that anyone with a PhD should at least understand the intuition behind a proposed method. The presentation is not friendly to an audience outside this particular domain—to say the least. The paper would benefit significantly from clearer explanations, concrete examples, graphical illustrations, and more explicit mathematical formulations.

---

> ### Author Response · Authors · 2025-11-28
> **Answer to reviewer 3dpa**
>
> We thank the reviewer for their detailed comments.
>
> We are actively working on a revised version of our manuscript, to address the structural issues brought up
> by the reviewers. More concretely, we focus on 1.) shortening introduction and background, dedicating more space
> and clarity to our method and contributions upfront. 2.) Review and clarify all mathematical notations. 3.) Adding details
> related to the description of our method.

---

### Official Review · Reviewer_3tMr · 2025-11-01

**Soundness:** 3
**Presentation:** 3
**Contribution:** 3
**Rating:** 4
**Confidence:** 2

**Summary:**

This paper addresses the limitations of Neural Processes (NPs) in time series forecasting, namely their inability to leverage context from multiple related time series and their simplistic handling of temporal structure. The authors propose a Conditional Neural Process (CNP) variant that treats each entire time series as a context example, aggregating information with a sequence-based encoder and Bayesian Aggregation. The approach is evaluated on challenging reinforcement learning-based simulation tasks, with extensive experiments comparing against strong baselines, including a transition-based CNP, a transformer-concat model, an oracle, and an uninformed baseline.

**Strengths:**

1. The central idea of treating an entire time series as a single context point within a meta-learning task is a conceptually elegant and powerful extension of the NP framework. This approach provides a more natural and structured way to handle collections of related time series, better aligning the model's inductive biases with the problem structure often found in real-world scenarios.
2. The paper is exceptionally logically structured and easy to follow. The authors clearly motivate the problem, formalize their approach, and situate it within the existing literature. The experimental design is sound, featuring appropriate and insightful baselines, which effectively establish upper and lower performance bounds and help to contextualize the results.

**Weaknesses:**

1. Experiments are confined to structured physics simulations (MuJoCo) and do not test noisy, heterogeneous real-world time series (e.g., finance, energy, retail), where task boundaries are less explicit. Claims of general applicability remain unverified on real data.

2. The method assumes all context series come from the same meta-task. The paper does not evaluate cases with irrelevant or out-of-distribution series, or show that Bayesian Aggregation effectively down-weights them, leaving reliability under realistic noise conditions uncertain.

**Questions:**

1. Real-world time series data often lack the clean "meta-task" structure present in the simulation environments. Could the authors comment on the potential applicability and challenges of their method on broader real-world time series benchmarks (e.g., from finance, energy, or demand forecasting)?

2. How does the model behave when the context set includes irrelevant or out-of-distribution series? Does Bayesian Aggregation actually down-weight such sequences via learned variances? Were any experiments conducted to explicitly validate this behavior?

---

> ### Author Response · Authors · 2025-11-28
> **Answer to reviewer 3tMr**
>
> We thank the reviewer for their valuable feedback and particularly appreciate their recognition of our method as a
> "conceptually elegant and powerful extension of the NP framework." We are also pleased that the reviewer found the
> paper's logical structure and experimental design to be sound.
>
> In the following we address the questions and weaknesses:
>
> - W1/Q1: We deliberately considered dynamical system benchmarks for our evaluation, since these simulations offer precise
> control over data generation and a clear meta-task structure with known meta-parameters. This allowed us to precisely
> evaluate the model's capability to infer concrete meta-tasks, as demonstrated in the paper. Regarding noise, we did
> incorporate heteroscedastic noise to the time series by adding action noise during trajectory generation.
>
>     While using datasets with clearly defined meta-task structures is a common and accepted practice in the meta-learning/neural process literature [1,2,3], we agree with the reviewer that incorporating further datasets is a valuable contribution. In particular, we fully agree that real-world data often lacks a clear meta-task structure. We are addressing this by conducting experiments with time series data from the UCR time series archive [4]. This collection consists of real-world time series from various domains. In these experiments, we are defining each domain as a separate meta-task. We will incorporate these results and an in-depth discussion of the findings into the updated manuscript (the experiments are currently running).
>
>     We would also be very grateful if the reviewer could suggest any further specific real-world time series datasets or benchmarks they believe would be particularly suitable for evaluating our method's performance and generalizability in practical settings.
>
> - W2/Q2:  While meta-learning commonly assumes that context data originates from the same task, using Bayesian Aggregation (BA) in our work is indeed intended to provide robustness. The original BA paper [1] demonstrates its ability to effectively down-weight
> less relevant context data by leveraging learned variances. However, we did not validate this specifically in our experiments.
> We consider incorporating specific ablation studies into the revised manuscript to empirically demonstrate how Bayesian Aggregation
> handles and down-weights irrelevant or out-of-distribution sequences.
>
> [1] https://openreview.net/pdf?id=ufZN2-aehFa
>
> [2] https://arxiv.org/abs/1910.13556
>
> [3] https://arxiv.org/abs/2303.14468
>
> [4] https://arxiv.org/abs/1810.07758

---

### Author Response · Authors · 2025-11-28
**General answer to all reviewers**

We would like to thank all reviewers for their effort in reviewing our paper and for their valuable feedback. We are very grateful for your generally positive remarks, which affirm our work as a
"conceptually elegant and powerful extension of the NP framework" (Reviewer 3tMr) that tackles the "fascinating and highly relevant" (reviewer Qe2b) as well as "practically meaningful" (reviewer oXKk) problem of meta-learning time series forecasting with few-shot examples
through a "clean theoretical idea" (Reviewer oXKk). We also appreciate the recognition that our "proposed model is simple and easy to understand" (Reviewer Qe2b), and our "experimental design is sound, featuring appropriate and insightful baselines" (reviewer 3tMr).

Your constructive comments and insightful suggestions have been instrumental for improving the paper. To address the most common points of feedback, we are expanding our experimental evaluation with real-world time series data (experiments are currently running),
significantly enhancing the clarity and structure of the manuscript, and strengthening the empirical validation of our method through further ablation studies. We are confident these revisions will substantially improve the paper.
We will upload an updated version of our manuscript as soon as possible.

Please find our detailed responses to your individual comments in separate answers to your reviews.

---

> ### Author Response · Authors · 2025-12-03
> **New revision**
>
> We have uploaded an improved version of our manuscript that now includes preliminary experimental results on real-world time series data, directly addressing the main issue raised by the reviewers. This new evaluation demonstrates that leveraging context time series, in addition to historical observations, significantly improves forecasting performance across diverse datasets compared to an uninformed baseline. Notably, our proposed method excels at effectively utilizing extensive contextual information, outperforming other contextual baselines. We will continue to refine the manuscript for the camera-ready version, incorporating further results and address any remaining minor issues.

---

### Meta-Review · Area_Chair_wYRN · 2026-01-01

**Summary:**

This paper introduces a novel Neural Processes method for time series forecasting by incorporating the idea of meta-learning, that treats related time series as conditionally independent context examples of a shared underlying data-generating process corresponding to a specific meta-task.

Despite an interesting idea, all reviewers consistently identify fundamental issues in (1) poor presentation/writing, (2) insufficient and unconvincing experimental validation, and (3) limited justification of novelty beyond existing Neural Processes methods.

Multiple reviewers explicitly state they could not fully understand the proposed method even after careful reading, which blocks proper evaluation of the technical contribution. On the other hand, several reviewers note strong assumptions and claims are not fully validated in the experiments.

Although the authors uploaded a revised version of the submission, the fundamental issues regarding paper quality still remain. Therefore, I recommend rejection.

**Reviewer Concerns:**

The presentation problem is addressed to some extent, but still it could be further improved.

Meanwhile, insufficient and unconvincing experimental validation and limited justification of some claims, are still outstanding.

**Reviewer Scores:**

Reviewer 3tMr would keep the score unchanged, rating 4 -> 4.

Reviewer 3dpa would keep the score unchanged, rating 2 -> 2.

Reviewer Qe2b would keep the score unchanged, rating 4 -> 4.

Reviewer oXKk would keep the score unchanged, rating 2 -> 2.

---

### Decision · Program_Chairs · 2026-01-26

Reject